# Failure to replicate the Aubert-Fleischl effect

**Björn Jörges** *, **Laurence R. Harris**

Center for Vision Research, York University, Keele Street, Toronto, Canada

* bjoern_joerges@hotmail.de

## Abstract

For many years the accepted wisdom in vision science has been that humans tend to underestimate the speed of an object when they pursue it with their gaze in comparison to when they fixate a spot in the visual scene and the object moves across the retina, an effect that is referred to as the Aubert-Fleischl phenomenon. However, experiments on the Aubert-Fleischl effect have generally employed a target moving in front of a blank background. For objects moving in the real world, relative motion between the object and the background provides additional cues, potentially allowing the visual system to compensate for the Aubert-Fleischl phenomenon. To test this hypothesis, we asked 50 participants to compare the speed of a single sphere to the speed of a sphere-cloud in a two-interval forced-choice task while they either followed the single sphere with their gaze or kept their eyes on a fixation cross. Stimuli were presented in virtual reality and the sphere's movement could occur either in a completely black environment with no relative motion cues present, or against a visible, textured background behind the moving sphere. The probe sphere-cloud always moved in front of the same background as the target. We found no evidence for an Aubert-Fleischl effect, i.e., no evidence for an underestimation of speed during pursuit relative to fixation, in either of the environments. Our results challenge the prevailing notion that object speed is underestimated when the object is pursued versus when fixation is maintained and highlights the necessity to take into account characteristics of the visual scene when it comes to the Aubert-Fleischl effect.

## Introduction

The Aubert-Fleischl phenomenon is the underestimation of the perceived speed of a target when an observer pursues it with their gaze compared to when viewing the same target moving across the retina during fixation [1,2,3,4,5,6,7,8,9]. In a related phenomenon, the Filehne illusion, non-fixated stationary objects can appear to move when their images are dragged across the retina during smooth pursuit of another target [10,6,11,8]. Differences between speed perception during pursuit and fixation arise from differences in the speed cues at play: while retinal cues are, in principle,

**Data availability statement:** All data, code and materials are made available in the following OSF repository: https://osf.io/xcaq3/. All OSF links mentioned in this manuscript refer to elements of this repository. Specifically, data and code can be found here (https://osf.io/xcaq3/files/xvrea), as well as the Unity projects (https://osf.io/6mkp8) used to generate the executables (https://osf.io/hqwra). The pre-registration can be accessed here (https://osf.io/byhg8). The original (pre-registered) data analysis script can be found here: https://osf.io/p9kwm. Please see the corresponding Supporting Information file for relevant information regarding pre-registration changes.

**Funding:** We thank the Canadian Space Agency for partial funding of this project (15ILSRA1-York). LRH was funded by Discovery Grant RGPIN-2020-06093 from the Natural Sciences and Engineering Research Council of Canada. BJ was funded by a Postdoctoral Transition Fellowship from the Centre for Vision Research at York University, Toronto. There was no additional external funding received for this study. The funders had no role in study design, data collection and analysis, decision to publish, or preparation of the manuscript.

**Competing interests:** The authors have declared that no competing interests exist.

sufficient to estimate the speed of a target moving across a fixated stationary scene, during pursuit the observer has to take into account both retinal and oculomotor cues to speed. The perceived speed of the eyes is thought to be underestimated based on oculomotor cues alone [2,12,6,11]. This then leads an observer to underestimate object speed when eye velocity is the only source of information (in the Aubert-Fleischl phenomenon) or to illusory motion of the scene (in the Filehne illusion).

The Aubert-Fleischl phenomenon is part of the reason why experiments on speed perception and time-to-contact estimation often require observers to keep their gaze on a fixation cross [13]. Embedding a tracked target into a visual scene provides additional cues to the speed of the eye – specifically relative motion between the target and the background – which should therefore make it easier to estimate the speed of an object accurately both during fixation and smooth pursuit. Credibility is lent to this idea by a study showing that relative motion supersedes absolute retinal motion for speed judgements under a variety of circumstances [14]. The – to our knowledge – most recent model on the Aubert-Fleischl effect [15] predicts the Aubert-Fleischl phenomenon via the observation that relative speed judgements become less precise, which in turn would make the slow-motion prior [16] affect perceived speed more strongly. Overall, this model predicts that adding more reliable motion cues should decrease or potentially even fully eliminate the Aubert-Fleischl effect. And in fact, some of the older studies on the Aubert-Fleischl effect [5], where some relative motion cues were present, failed to find a difference between fixation and pursuit. Even Aubert himself was aware of the fact that relative motion between a moving target and a static target might affect perceived speed but he did not specifically address to what extent the presence of a static object or a background might impact the Aubert-Fleischl effect (1, 2). If the Aubert-Fleischl effect is, indeed, not present under such circumstances, this would provide researchers with more flexibility to allow their participants to pursue targets, have them keep their gaze on a fixation cross, or allow natural free viewing. This project aims to provide conclusive evidence by first attempting to reproduce the original Aubert-Fleischl effect in a VR set-up, and then making it disappear by adding relative motion cues between the target and background. Another practical consideration for research practice is that most of the studies into the Aubert-Fleischl effect (with some notable exceptions; [17,6,7]) did not use individual targets for speed judgements, but rather gratings or other repeating patterns [2,18,3,4,19,15,5,20]. Given some studies [19,15,21] investigating the interactions between stimulus characteristics and the Aubert-Fleischl effect, and given that many studies in areas such as time-to-contact estimation and speed perception use single targets rather than target patterns and gratings, it seems important to determine whether the Aubert-Fleischl effect actually applies to individual stimuli.

Our two hypotheses are:

1. The Aubert-Fleischl effect can occur while tracking a target in a VR display when all relative motion cues between background and target are removed.

2. The Aubert-Fleischl effect disappears in the presence of relative motion cues between background and target (as in most VR applications).

## Methods

### Participants

We collected data until 50 participants had successfully completed the initial eye movement training and the two-interval forced-choice task training; this was achieved after testing a total of 69 participants (23 men, mean age = 22.5 years; 41 women, mean age = 21.7 years; 4 participants declined to answer, mean age = 20.9 years). All participants had normal or corrected-to-normal vision. Participants were recruited from the York University undergraduate participant pool and were rewarded with course credit. The study was approved by the York University Office of Research Ethics (ORE) and was conducted in accordance with the principles of the Declaration of Helsinki. Written informed consent was obtained.

### Apparatus

Our stimuli were presented in an HTC VIVE Pro Eye head-mounted device with a 90 Hz refresh rate, a diagonal field of view of 110° and a resolution of 1080 x 1200 pixels. Participants responded with the HTC VIVE Pro Eye controllers. The VIVE Pro Eye records eye-movements at 120 Hz (binocular) with a spatial accuracy of 0.5° to 1.1°. The program used to collect data was built in Unity (v. 2020.3.14f). For the "no relative motion" condition (see below), we attached Kodak ND9 filters to the lenses of a VIVE Pro Eye to make sure that the edge between the lenses and the plastic fixture were not visible and did not provide any relative motion cues. This was necessary because even when no ambient lighting is simulated via Unity, the VIVE Pro Eye's pixels never fully turn off, making for a distinctly visible visual edge even at the lowest brightness settings. The VIVE Pro Eye eye-tracker was, unfortunately, not able to record eye movements under these conditions; see below for measures that we put in place to ensure that participants followed gaze instructions even when we could not verify their behavior.

### Stimulus

In a two-alternative forced-choice task, participants judged the speed of a red target sphere which moved from either left to right or from right to left, against the speed of a sphere cloud that moved in the same direction as the target and that could be presented either before or after the single red sphere. The reason we chose a sphere cloud as our comparison stimulus is that comparing the speed of a single sphere to the speed of another single sphere would allow the observer to use either the ball's travel distance (when presentation time was matched) or presentation time (when traveled distance was matched) rather than its speed. In this sense, the sphere cloud is akin to a 3D moving Gabor patch in that it allows us to decouple speed, distance, and presentation time. See Fig 2B for the temporal sequence of presentation within each trial.

The sphere was simulated as being 7.5 m in front of the observer with a diameter of 0.2 m (corresponding to a visual angle of 1.47° at the midpoint of its trajectory and 1.53° at its right and left-most positions). The wall was simulated as being 0.5 m behind the sphere i.e., 8 m in front of the observer. The sphere was visible for 1.1 s in total, which was divided in three phases: an acceleration or entrainment phase (0.3 s), a constant speed phase (0.5 s) and a deceleration phase (0.3s). Given that the Aubert-Fleischl phenomenon only occurs during smooth pursuit eye-movements, we added an initial acceleration phase to make sure that, in the pursuit condition, no catch-up saccades were executed during the constant speed phase. No such phase was necessary for the sphere cloud as participants were always asked to fixate during that part of the trajectory.

The sphere appeared either to the left or to the right of the observer's visual straight-ahead. The initial position ($x_0$) was chosen such that the overall trajectory, consisting of the space travelled during acceleration ($x_{acc}$; Equation 2), constant speed ($x_{cons}$; Equation 3) and deceleration ($x_{dec}$; Equation 4) was centered in front of the observer (Equation 1). The acceleration ($a_{acc}$) was chosen such that the sphere accelerated to its final speed ($v_{max}$; 2, 4, or 6 m/s, which amounted to angular speeds of about 15°/s, 30°/s and 45°/s relative to the observer respectively) over 0.3 s (Equation 5)..

$$x_0 = -(x_{acc} + x_{cons} + x_{dec})/2 \tag{1}$$

$$x_{acc} = \frac{(a_{acc} * t_{acc}^2)}{2} \tag{2}$$

$$x_{cons} = v_{max} * t_{const} \tag{3}$$

$$x_{dec} = \frac{(-a_{dec} * t_{acc}^2)}{2} \tag{4}$$

$$a_{acc} = \frac{v_{max}}{t_{acc}} \tag{5}$$

with $t_{acc} = 0.3s$

$$a_{dec} = -a_{acc} \tag{6}$$

The sphere cloud was presented at the same distance (7.5m) and had a width of 2.5 m (spanning a retinal angle of 38°). Each element of the cloud had the same size, shape and color as the single sphere, was moving in the same direction as the single sphere on the trial. Each individual element of the cloud appeared on one side and moved across the 2.5 m width over a time that was determined by its speed. The element then disappeared. New sphere cloud elements were generated continuously such that around 20 were visible at any given time. The sphere cloud's speed was governed by a PEST staircase [22], one starting 30% above the actual speed of the target and one starting 30% below the actual speed, for a total of two staircases per speed per condition. To restrict the total length of the experiment, each staircase was terminated after 15 trials.

Participants were asked to either:

(1) fixate a fixation cross presented at the same distance as the spheres but 0.5 m (3.8°) below the height at which the single sphere was presented (Fixation condition),

(2) or follow the sphere with their gaze (Pursuit condition).

Participants were asked to always keep their gaze on the fixation cross during presentation of the sphere cloud. The stimuli were presented either:

(1) In front of a textured wall backdrop that provided relative motion cues between the target sphere and the background. This was part of a larger virtual 3D environment in which participants were immersed (Relative Motion condition, see Fig 1A and B; see also this video https://osf.io/v23p5)

(2) in a uniformly black scene that did not provide any relative motion cues (No Relative Motion condition, see Fig 1C and D; see also this video https://osf.io/sjb3k),

**Eye movements.** We recorded the participants' eye-movements while the sphere or the sphere cloud was visible. These recordings were used later to exclude participants that consistently did not follow the instructions, or individual trials where an otherwise acceptable participant failed to follow the instructions. Please note that eye-movements could not be

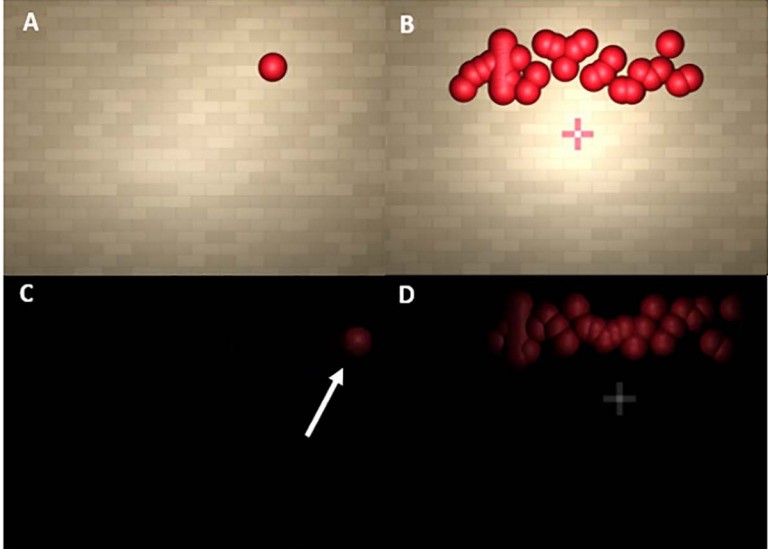

**Fig 1. Screenshots from the experiment in the Relative Motion condition (A, B) and the No Relative Motion condition (C, D), for when the single sphere was on screen (A, C) or while the sphere cloud was on screen (B, D).** The white arrow in Panel C is meant to make the position of the sphere more obvious to the reader and was not present in the experiment.

recorded for the No Relative Motion condition because the eye cameras required more light to function than was available within the headset during this condition (see below for more information on inclusion criteria for participants and individual trials).

Since managing gaze behavior and our speed discrimination task at the same time is a fairly demanding task, either the single sphere or the fixation point was flashed white intermittently to indicate precisely where participants had to direct their gaze for the first five trials of each block.

The programs used to present stimuli, and the corresponding Unity project can be downloaded from https://osf.io/xcaq3/

### Procedure

In total, participants completed 6 stages of the experiment: An eye-movement training session, an eye-movement test, a two-interval forced-choice task training session, the first experimental block (Relative Motion or No Relative Motion, with either Fixation or Pursuit first), another eye-movement assessment, and finally the second experimental block (Relative Motion or No Relative Motion, with either Fixation or Pursuit first). See Fig 2A for a schematic of the sequence of tasks.

**Eye-movement training.** Participants completed one run of 25 trials with a visible backdrop. Participants were instructed to always follow the single sphere with their gaze and keep their eyes on the fixation cross while the sphere cloud was present. They were also instructed to focus mostly on their gaze behavior and, if necessary, respond randomly to the speed judgement task. After each trial, they were given feedback on whether their gaze behavior matched the criteria established as sufficient performance: their eye speed had to be at least 30% of the sphere's speed when the sphere was present, and it had to be below 30% of the sphere's speed when the sphere cloud was present. Trials were repeated when the gaze behavior did not match the instructions. Throughout this training, the experimenter explained the expected eye behavior as many times as necessary.

**Eye-movement assessment.** After finishing the eye-movement training, we assessed participants' eye behavior in a 50-trial test (again, with a visible backdrop because eye-movements could not be recorded in the dark condition). Before

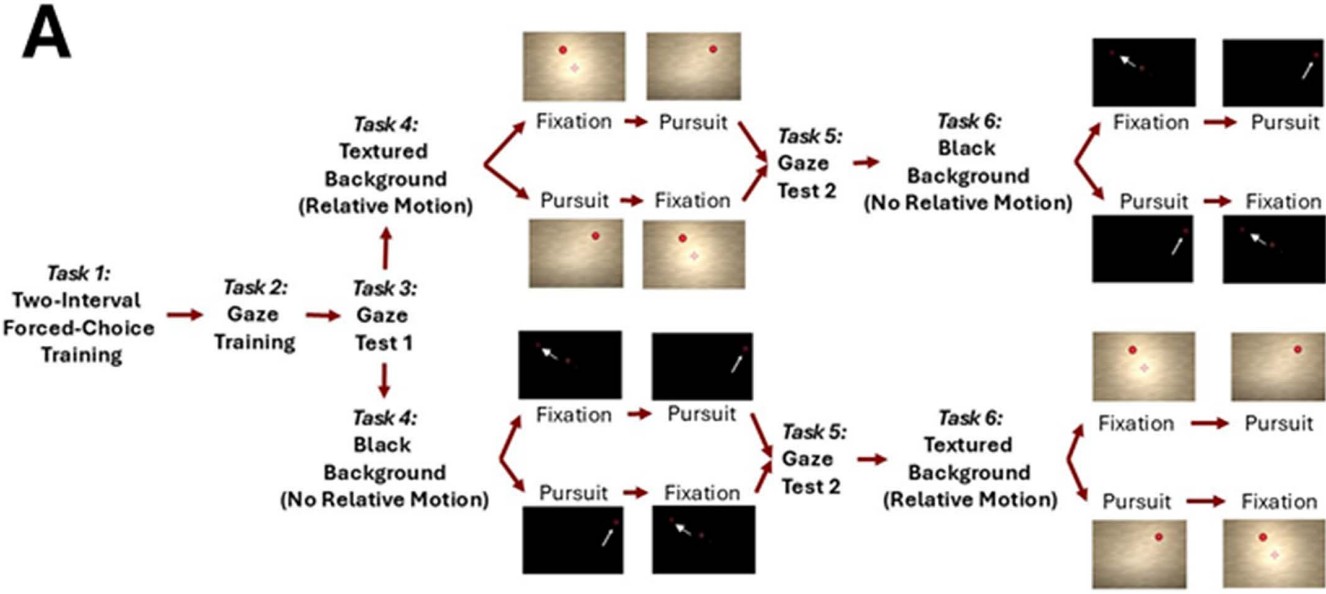

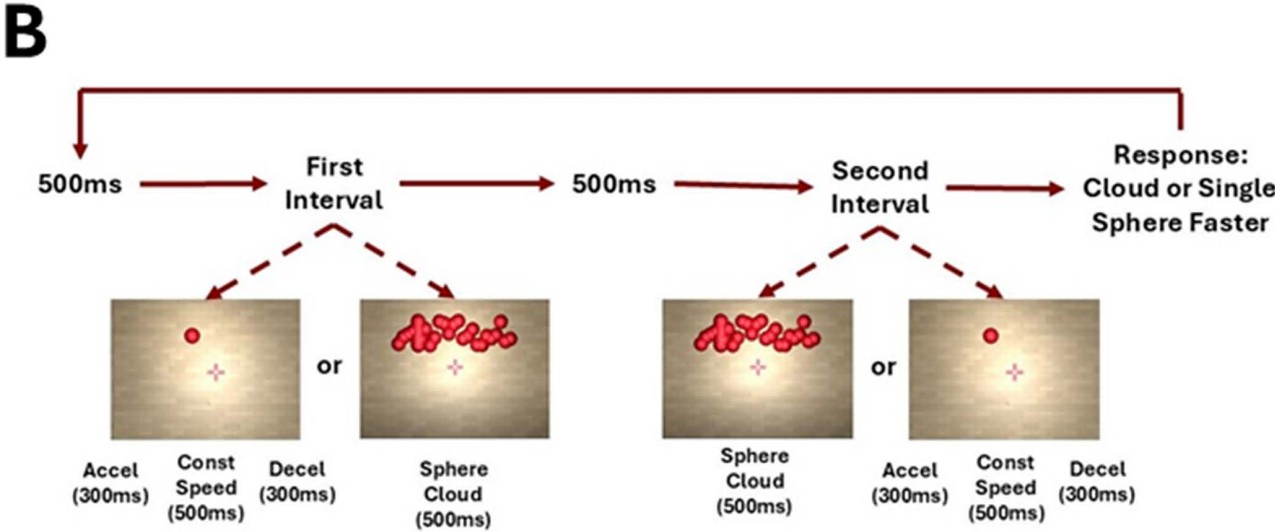

**Fig 2. A. Sequence of tasks as described in the Procedure section. B.** Sequence of presentation along with the respective timings within each trial in the main experimental conditions.

each trial, participants received instructions as to whether they were supposed to follow the sphere with their eyes or keep their gaze on a fixation cross. At the end of the 50-trial run, the program output the participants median gaze speed throughout the 50 trials, separately for Pursuit and Fixation trials, and separately for those parts of the trial where the sphere was present and those parts where the sphere cloud was present. The median gaze speed had to be at least 30% of the presented speed in the Pursuit condition while the sphere was visible, and below 30% of the presented sphere speed for the Fixation condition, as well as for those parts of the trial where the sphere was presented. Participants who

did not satisfy these criteria received another explanation of the expected gaze behavior and underwent a second round of the test. If they still did not meet the criteria, they were sent home. This was the case for 11 participants out of 61 participants who started the experiment.

**Two-Interval forced choice task training.**  After an explanation of the task, participants were asked to complete a short training which consisted in a reduced (30 trial) version of the main experiment; the speed of the sphere was set to 5 m/s and only one staircase (where the speed of the sphere cloud started 30% above the speed of the single sphere) with a maximum of 25 trials presented. The training was done in the background condition the respective participant started in, i.e., participants who started with the Relative Motion condition did their training in the Relative Motion condition and participants who started with the No Relative Motion condition did their training in the No Relative Motion condition. In the training, the fixation cross was absent, and they were allowed to view the stimulus freely. Participants were only asked to continue with the main experiment if the last step size in the staircase was below 0.3 m/s, indicating that they had an acceptable understanding of task and controls. If participants failed to meet this criterion, they received another explanation of the task and were asked to repeat the training. No participants failed the training a second time.

**First experimental block.**  Participants then proceeded to the first experimental block. Half of the participants started with the Relative Motion condition and half started with the No Relative Motion condition. Within each relative motion condition, half of the participants started with the Fixation condition and half started with the Pursuit condition. Within each gaze condition, we consistently presented the single sphere during the first half, and the sphere cloud first during the second half, or the other way around. Before the start of the No Relative Motion condition, participants sat in the dark for about one minute such that the experiment took place after dark adaptation plateaued.

**Second eye-movement assessment.**  After the first experimental block, participants completed another eye-movement test to assess whether they were still following gaze instructions. No participants failed this test.

**Second Experimental Block.**  Participants then completed the experimental block they had not completed before (Relative Motion or No Relative Motion). The gaze instructions switched, as before, half-way through the block.

Between the eye-movement training and the two eye-movement assessments, we were confident that all included participant displayed the expected eye-movement behavior across both background conditions.

## Predictions

We expected the speed of the sphere to be underestimated when the participant pursued the target with their gaze in comparison to when they maintained fixation in the dark. We expected this difference to disappear when the target moved against the textured background.

## Data analysis

The code used for the analysis along with the data can be downloaded from Open Science Foundation: https://osf.io/xcaq3/files/xvrea

**Preprocessing of eye movement data.**  We recorded gaze data in world coordinates in the plane in which the target was presented. Since our target was only moving laterally, we only analyzed the lateral eye speed component. Visual inspection of a random sample of gaze traces confirmed that the vertical component was largely constant across the trial. We smoothed the horizontal component using a Gaussian kernel and computed gaze speed on a frame-to-frame basis. All frames where the eye speed was higher than twice the target speed were marked as saccades (see Figure A5 in the S1 Appendix for a sample of eye traces with saccades marked up of 15 trials from one random participant as well as Figure A6 for the distributions of saccades for each participant in the Pursuit-No Relative Motion condition). We then removed these frames and computed the mean gaze speed across the remaining frames for further analysis.

**Outlier analysis: Step 1.** A trial was marked as "instructions not followed" when the average gaze speed was < 50% of the target speed in the Pursuit condition (see Figure A1 in the S1 Appendix), or > 33% of the target speed in the Fixation condition (see Figure A2 in the S1 Appendix; see also Figure A3 for the distributions of mean eye speeds for each participant while the sphere cloud was present). While comparing the gaze speed to the target speed, we accounted for the 50 ms it takes the VIVE Pro Eye to make the eye-tracking data available to the system [23]. We also excluded all trials where we identified more than 3 saccades. Based on this criterion, we excluded conditions (defined as combinations of the two Gaze conditions and three target speeds) that, out of 30 trials for any given speed and condition, had 20 or more trials marked as "instructions not followed". We further excluded any participant for who more than one of the conditions was excluded using this criterion, under the assumption that we were not confident enough that their eye behavior matched the instructions in the No Relative Motion condition for which eye data could not be collected. Four participants (n = 4) were wholly excluded based on these criteria, and 3 conditions out of a total of 276 (1.1%) from the remaining 46 participants were removed. Finally, out of all remaining trials (8190), we excluded 1400 trials (17.1%). Given that eye-tracking data was not available for the No Relative Motion condition (see apparatus), no conditions or trials were excluded here.

**Fitting psychometric functions.** We then fitted cumulative Gaussian functions separately for each target speed, condition and participant by means of direct likelihood maximization [24] as implemented in the quickpsy package [25] for R (R Core [26]). We then extracted the PSEs (i.e., the 50% values) as a measure of accuracy and the JNDs (i.e., the 84.1% discrimination thresholds) as a measure of precision. Fig 3 illustrates this process for one condition with a single sphere speed of 4 m/s.

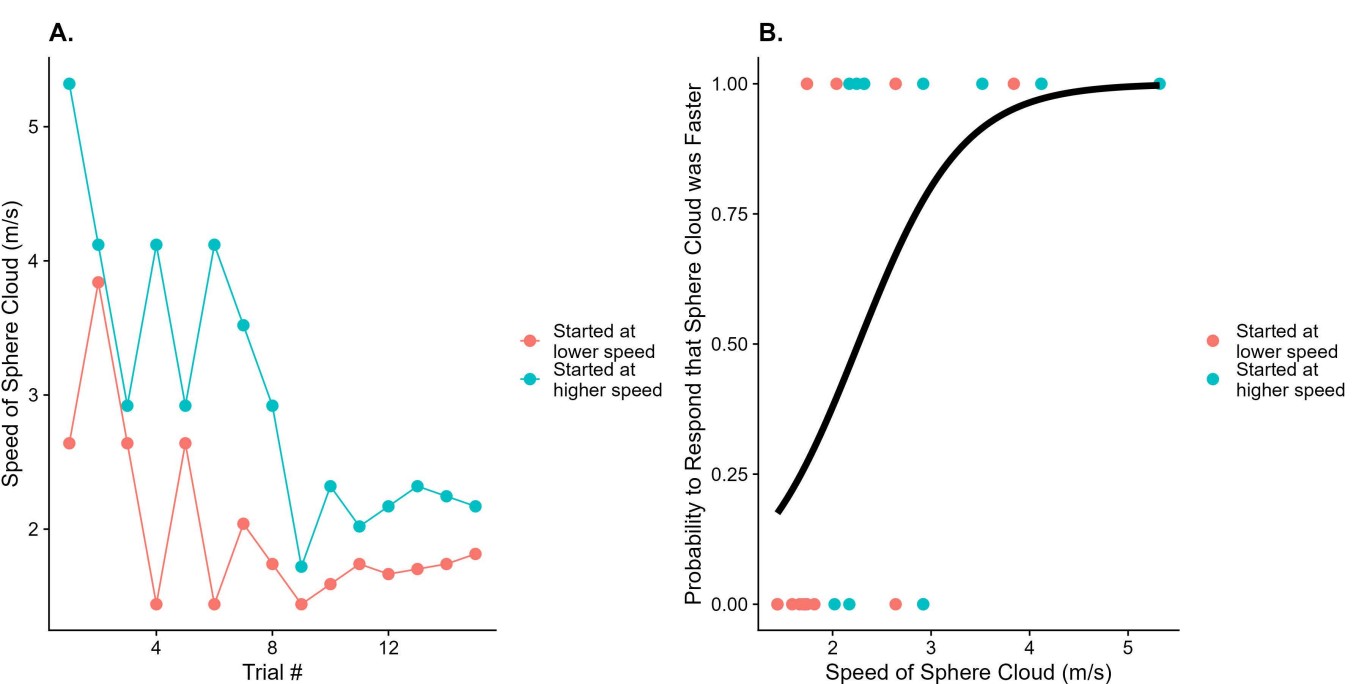

**Fig 3. A. Depiction of how the presented speed of the sphere cloud (y axis) developed over the course of the experiment (trial number on the x axis), color-coded for whether the speed of the sphere cloud started out being higher than the speed of the single sphere (blue) or lower (red). This was for a single sphere speed of 4 m/s. B.** Example of a fitted psychometric function (solid black line) for the same condition depicted in **A.** The probability for the participant to respond that the sphere cloud was faster than the single sphere (y axis) is plotted against the speed of the sphere cloud (x axis). Individual responses for trials are shown as dots; 1 equals the response "sphere cloud was faster" and 0 denotes the response "single sphere was faster", color-coded as in **A.** The Point of Subjective Equality (PSE) is defined as the speed of the sphere cloud at the 50% threshold, indicated by the intersecting dashed lines.

**Outlier Analysis: Step 2.** Once we fitted the psychometric functions, we removed those conditions for which impossible or unreasonably extreme PSEs or JNDs were obtained: any conditions where PSEs or JNDs below 0 were obtained were excluded, as well as any conditions where PSEs more than 3 times higher than the presented speed were fitted. This led to the exclusion of 100 conditions (out of a total of 549 remaining ones), or 18.2%.

**Main statistical analysis.** Since we expected to detect a difference between Fixation and Pursuit only in the No Relative Motion condition, we separated the Relative Motion condition and the No Relative Motion condition for analysis. Since frequentist approaches are generally ill-suited to quantify evidence for the absence of an effect (such as the absence of a difference between Fixation and Pursuit in the Relative Motion condition), we relied on a Bayesian approach for all analyses.

To test for a difference between Fixation and Pursuit (separately for the Relative Motion and the No Relative Motion conditions), we used Bayesian Linear Mixed Modelling as implemented in the brms package [27] for R to test for differences between the experimental conditions. We first fitted a model with the PSEs as the dependent variable, the gaze condition (Fixation vs. Pursuit; categorical variable) as a fixed effect, as well as random intercepts and random slopes for target speed (in m/s) for the grouping variable Participant. In [28], this model reads as follows:

$$PSE \sim Gaze\ Condition + (1 + Speed_{Sphere} \mid Participant) \tag{7}$$

We then used the hypothesis() function from the brms package to determine the Bayes Factor for a difference between Fixation and Pursuit. As per our predictions (see Predictions), the Bayes Factor should indicate evidence for the absence of a difference between Fixation and Pursuit in the Relative Motion condition, while indicating evidence for PSEs being underestimated for Pursuit relative to Fixation in the No Relative Motion condition (the classic Aubert-Fleischl effect).

## Results

Fig 4 provides an illustration of the PSEs for all conditions, and Table A1 in S3 Appendix provides an at-a-glance view of all fitted fixed effects coefficients for all statistical models. The preregistration can be found on OSF (https://osf.io/byhg8). Please note that substantial changes have been made to the original analysis in response to reviewer feedback. A list of these changes can be found in the Open Science statement.

### Pre-registered analysis on the Aubert-Fleischl effect without relative motion

Contrary to our prediction, we found weak evidence in favor of the hypothesis that PSEs were equal between Fixation and Pursuit (Bayes Factor = 2.2). Numerically, PSEs were 0.1 m/s (95% credible interval = [−0.17 m/s; 0.36 m/s]) higher in the Pursuit condition than in the Fixation condition.

### Pre-registered analysis on the Aubert-Fleischl effect with relative motion

Regarding our hypothesis that there would be no difference between Pursuit and Fixation in the Relative Motion condition, we found weak evidence in favor of the absence of an effect (Bayes Factor = 2.1), with numerically higher PSEs in the Pursuit condition than in the Fixation condition (by 0.11 m/s, 95% Credible Interval = [−0.11 m/s; 0.33 m/s]).

These results were highly unexpected. Not only did we not find confirmation of our main hypothesis (namely that speed would be underestimated during pursuit relative to fixation, but only when no relative motion was present). Rather, we found that the perceived speed was numerically higher in the pursuit condition in both the Relative Motion condition and the No Relative Motion condition – which would constitute a *reverse* Aubert-Fleischl effect. We therefore also performed a (Bayesian) Generalized Linear Mixed Model analysis, which was previously found to boast higher statistical power [29] and which we had initially only pre-registered out of methodological interest (see S1 Appendix), in order to verify whether these unexpected findings would be further supported.

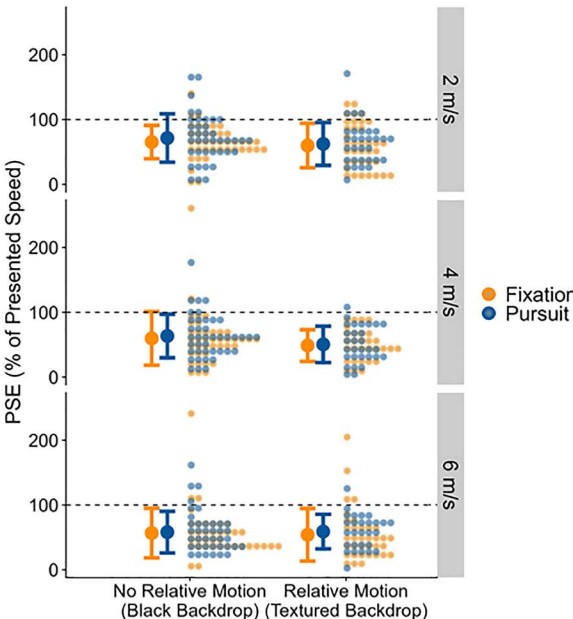

**Fig 4. Full distributions of the fitted PSEs (y axis) for the two relative motion conditions (x axis), eye behavior conditions (color-coded) and sphere speeds (panels), along with the means (bold dots) and +−1 standard deviation (error bars) across all data points in that condition. The dashed line represents perfect performance, i.e., a veridical perception of ball speeds. The panels represent the different ball speeds.**

### Secondary pre-registered generalized linear mixed model analysis on the Aubert-Fleischl effect without relative motion

This analysis provided moderate evidence that Pursuit elicited the same likelihood of responding "cloud faster" as Fixation (Bayes Factor = 5.1). Rather than reporting (hard to interpret) regression coefficients, we used the method outlined by Moscatelli et al. [29] to convert these regression coefficients into PSEs. This showed that PSEs were numerically higher in the Pursuit condition than in the Fixation condition (by 0.09 m/s, 95% Credible Interval = [−0.18 m/s; 0.38 m/s]).

### Secondary pre-registered Bayesian generalized linear mixed model analysis on the Aubert-Fleischl effect with relative motion

For the Relative Motion condition, we found weak evidence for Pursuit eliciting higher PSEs than Fixation (Bayes Factor = 3.02). Numerically, the PSEs were higher for Pursuit than for Fixation (by 0.16, 95% Credible Interval = [0.06 m/s; 0.43 m/s]).

These analyses were in line with the main pre-registered analyses reported above but provided a higher degree of statistical power, as evidenced by the smaller credible (BGLMM) intervals (see Fig 5) they provide on their parameter estimates.

Given these surprising results, we conducted two additional, **exploratory** (Generalized Linear Mixed Model-based) analyses to investigate whether pursuit really increased perceived speed: for the first one, we assessed whether trials where the participants' eyes moved faster were more likely to lead to responses that the sphere was faster, i.e., did faster eye motion on a trial-by-trial basis bias participants to perceive the sphere as faster? To this end, we added the mean eye speed (in m/s in the target plane) for each trial as a fixed effect to the Generalized Linear Mixed Model used above:

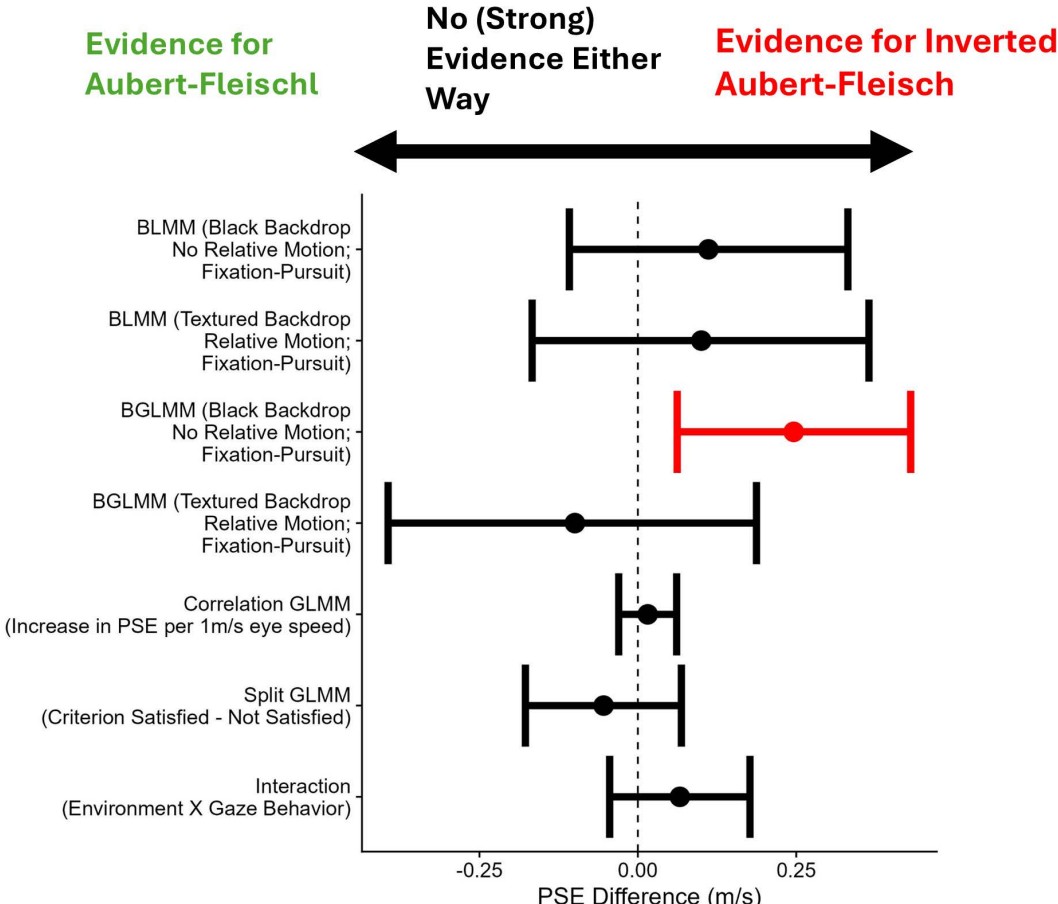

**Fig 5. Difference contrasts between Fixation and Pursuit (1-4), regression coefficient for the association between the eye speed and the likelihood to respond: "sphere faster", (5) and the difference contrast between those trials that satisfied the eye behavior criterion versus those trials that did not.** The error bars indicate the 95% confidence intervals (1, 3, 5, 6) or the 95% credible intervals (2, 4). The vertical line indicates no effect, while values below zero (to the left of the line) are consistent with the Aubert-Fleischl effect and values above zero (to the right of the dashed line) are consistent with an inverted Aubert-Fleischl effect (where object speed is overestimated in the presence of pursuit rather than underestimated). Statistical tests that provide evidence for an inverted Aubert-Fleischl effect are colored in red, while those colored in black do not provide (strong) evidence either way.

$$Response \sim Speed_{Eyes} + Number\ of\ Saccades + Speed_{Cloud} + Speed_{Sphere}$$
$$+ (1 + Number\ of\ Saccades + Speed_{Cloud} + Speed_{Sphere} \mid Participant) \tag{8}$$

For the second, conceptually similar analysis, we labelled trials as either "eye speed above criterion" (i.e., above 50% of the speed of the sphere) or "eye speed below criterion" (i.e., below 50% of the speed of the sphere), thus splitting the data set in two parts. We then compared whether performance differed between the two types of trials:

$$Response \sim Trial\ Type + Number\ of\ Saccades + Speed_{Cloud} + Speed_{Sphere}$$
$$+ (1 + Number\ of\ Saccades + Speed_{Cloud} + Speed_{Sphere} \mid Participant) \tag{9}$$

We conducted both of these analyses over all pursuit trials from the relative motion condition, including those trials where the eye criterion was not satisfied.

**Exploratory Analysis: were trials with a higher gaze speed more likely to lead to "sphere is faster" responses?**  We found strong evidence that gaze did not affect perceived speed (Bayes Factor = 25.5).

**Exploratory Analysis: were trials that satisfy the eye behavior criteria more likely to lead to "sphere is faster" responses?**  This analysis provided moderate evidence that Pursuit trials where the eye criterion was met did not elicit different PSEs than trials where it wasn't met (Bayes Factor = 5.1).

**Exploratory Analysis: Is there any interaction between Gaze condition and Relative Motion condition?**  Lastly, we also included an analysis to directly test whether there was an interaction between Gaze condition and Relative Motion condition to support our pre-registered split analyses above. We fitted a Bayesian Generalized Linear Mixed Model as described in Appendix A S1 Appendix, except that we added Relative Motion condition and the interaction between Gaze condition and Relative Motion condition as fixed effects. The full model structure was the following:

$$Response \sim Gaze\ Condition * Relative\ Motion\ Condition + Number\ of\ Saccades + Speed_{Cloud}$$
$$+ Speed_{Sphere} + (1 + Number\ of\ Saccades + Speed_{Cloud} + Speed_{Sphere} \mid Participant) \tag{10}$$

This analysis provided strong evidence that there was no interaction between Gaze condition and Relative Motion condition (Bayes Factor = 7.92). Fig 5 shows the parameter estimates of all relevant statistical comparisons reported, along with their 95% confidence (for the frequentist analyses) or credible (for the Bayesian analyses) intervals.

**Exploratory Analysis: did precision vary between fixation and pursuit?**  Finally, as a last exploratory analysis, we assessed whether any differences in precision were appreciable between Fixation and Pursuit. Precision was previously shown to be lower during pursuit, which 14[14] used, in conjunction with a slow motion prior [16], to explain the Aubert-Fleischl effect. To test whether our data were consistent with this model, we fitted a Bayesian Linear Mixed Model similar to our accuracy analyses, where we used the fitted JNDs of the psychometric functions as dependent variable, Relative Motion (Relative Motion vs. No Relative Motion) and Gaze condition (Fixation versus Pursuit) as fixed effects and the same random effects as above:

$$JND \sim Gaze\ Condition + Relative\ Motion + (1 + Speed_{Sphere} \mid Participant) \tag{11}$$

We found strong evidence for JNDs being higher for Pursuit than for Fixation (by 0.27 m/s, 95% Credible Interval = [−0.05 m/s; 0.59 m/s], Bayes Factor = 10.83). The JNDs are visualized in Fig 6.

Table A1 in S3 Appendix provides an overview over all difference contrasts considered in the models along with 95% credible intervals.

In S4 Appendix, we provide several figures that detail participants' gaze behavior on a condition-by-condition basis.

## Discussion

Contrary to our hypotheses, and contrary to what has generally been reported in the literature [1,2,18,3,30,31,17,4,19,15,32,11,7,20], we found evidence that perceived speed is unaffected when gaze follows an object in comparison to when gaze is kept on a fixation cross.

Comparable results have been previously reported by Gibson et al. [5], and some conditions reported by Dichgans et al. [31], and Freeman & Banks [19] but do seem not to have made it into general awareness. Gibson himself [5] showed participants two downward moving bands with dots on them. In the pursuit condition, they were instructed to follow one dot on one band to get an impression of its speed, then look over to the other band and adjust its speed while fixating a black fixation square that was positioned next to the band. In the fixation condition, they executed the same task but fixated black fixation squares for both stimuli. In both conditions, they were allowed to go back and forth between the two stimuli until they felt they had matched the speeds correctly. While speed was numerically overestimated in the fixation

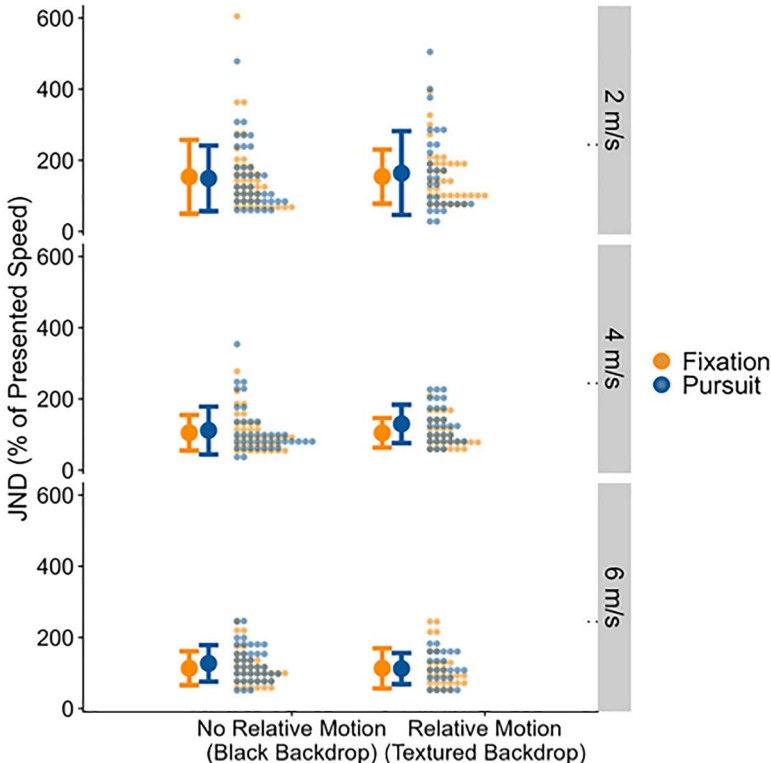

**Fig 6. Full distributions of the fitted JNDs (y axis) for the two relative motion conditions (x axis), eye behavior conditions (color-coded) and sphere speeds (panels), along with the means (bold dots) and +−1 standard deviation (error bars) across all data points in that condition.** The panels represent the different ball speeds.

condition and underestimated in the pursuit condition, this difference did not rise to significance. While the single edge condition from Dichgans et al. [31] is in some ways similar to our experiment, it is important to point out the differences: specifically, we were simulating a real target rather than moving one edge across the screen, further enhanced by the spatial separation between target and background. It is also important to note that several studies using a single target (rather than a repeating pattern) found the Aubert-Fleischl effect (e.g., the initial studies conducted by [1,2,4]. Our study therefore provides important information beyond Dichgans et al. [31].

A further complication in the interpretation of our results is that we found evidence for decreased precision in the Pursuit condition. In line with the model suggested by Freeman et al. [15], this should then causally lead to lower PSEs for which we do not find evidence. Specifically, this Bayesian model is based on the assumption that the perception of object speed becomes less precise during pursuit, which would make speed perception more susceptible to the slow motion prior [16] and thus decrease PSEs. Along with our data, which did demonstrate that JNDs can be increased without an accompanying decrease in PSEs, a recent study by de la Malla et al. (2022) [33] further complicates the picture by demonstrating that perceptual precision can even be *increased* during pursuit.

Another explanation might be found when examining the impact of spatial frequency on the Aubert-Fleischl effect. Dichgans et al. [31] found no differences between fixation and pursuit in a design where a single edge moved across a screen. Only for higher spatial frequencies, i.e., when a pattern of several vertical stripes was presented, did they find the expected Aubert-Fleischl effect. Drawing on these ideas about the effect of spatial frequency content, Freeman & Banks [19] proposed a model that predicted a reversal of the Aubert-Fleischl effect at low spatial frequencies and supported this

reversal experimentally. This is compatible with our results: the spatial frequency associated with our 1.5 deg wide stimulus would be primarily around 0.3 c/deg. The spatial frequency of the whole scene was much higher in the Relative Motion condition (where a tiled wall was simulated behind the stimulus), but the stimulus sphere itself had the same low spatial frequency. The Freeman & Banks model, however, assumes that all retinal motion is caused by the target, which makes it not directly applicable to our stimulus.

However, it is worth considering the assumptions of their model, namely that the perceived head-centric speed of a moving object can be calculated as a simple sum of perceived eye speed and perceived retinal image velocity. Contrary to many accounts of the Aubert-Fleischl phenomenon, which attribute it to an underestimation of eye speed [2,12,6,11], they posit that the phenomenon is instead caused by a misestimation of retinal image velocity (whose perception has been shown to be influenced by factors such as spatial frequency of the stimulus). This is, to some extent, in agreement with the fact that neither of our analyses show a significant difference between fixation and pursuit in the No Relative Motion condition: here, the difference in retinal image velocity between fixation and pursuit is negligible. On the flipside, in the Relative Motion condition, retinal image speeds are vastly higher during pursuit than during fixation because of the stationary background being dragged across the retina. This retinal motion goes in the opposite direction of the world-centric direction of the sphere. If some of this retinal motion were to be misattributed – perhaps as world motion, or as the consequence of bodily self-motion – the speed of the object might be slightly overestimated, potentially accounting for our data.

## Limitations

There are several potential limitations to this study, the most important one being our inability to verify participants' eye movements in the No Relative Motion condition. While we mitigated this shortcoming of our technology by having participants complete eye movement tests before each experimental block, the question remains whether the visual differences between the Relative Motion and the No Relative Motion conditions might lead to systematic changes in eye movements. Lower contrast between target and background, for example, may lead to less successful ocular pursuit, which in turn could lead to a decrease in the Aubert-Fleischl effect as reported in the literature. However, such effects of the stimulus on pursuit or saccades are limited to low contrasts close to threshold – much lower than those employed in our experiment [34]. It seems therefore highly unlikely that differences in gaze behavior would be the reason for the absence of an Aubert-Fleischl effect in our experiment.

Second, our method of entraining smooth pursuit (an acceleration phase in the presented stimulus) entailed slower speeds in the beginning of each trajectory. and some studies have found early motion information to be privileged over later information [35]. In the fixation condition, the acceleration phase was presented in the periphery while these slower parts of the trajectory would be presented much closer to central vision. While this is an aspect of our experiment that we did not control for, peripheral vision tends to be about as sensitive to speed information as central vision (e.g., [36]), making it unlikely as a confound.

## Conclusions

Contrary to our predictions, we did not find large differences in perceived object speed during fixation in comparison to when the object was pursued. While our data do not support a Bayesian model [15] based on the idea that decreased precision during pursuit could make judgements more vulnerable to a slow motion prior, another model [19] positing that object speed is judged accurately during pursuit but is more susceptible to stimulus properties like spatial frequency during fixation, provides a promising starting point for further investigation.

Overall, our results from a large sample of 50 participants suggest that the Aubert-Fleischl effect is unlikely to affect perceived speed drastically in any context where an observer pursues a single moving object with their gaze. We conclude that it is important for research and other VR applications to appreciate that the Aubert-Fleischl effect is highly context dependent.

## Supporting information

**S1 Appendix. Appendix A: Secondary Statistical Analysis using Generalized Linear Mixed Modelling.** This Appendix provides additional information on a secondary analysis we conducted using (Bayesian) Generalized Linear Mixed Modelling rather than the main (Bayesian) Linear Mixed Modeling-based analysis described in the main text.
(PDF)

**S2 Appendix. Appendix B: Plots of eye behavior per participant.** Appendix B provides additional graphics on important eye-behavior metrics such as gaze gains and saccades.
(PDF)

**S3 Appendix. Appendix C: Summary of all difference contrasts of all variables in all statistical models fitted in this paper.** Appendix C provides a detailed breakdown of all difference contrasts relating to all variables for all statistical models reported in this paper, including 95% credible intervals and Bayes Factors where relevant.
(PDF)

**S4 Appendix. Appendix D: Supplementary Analysis of Gaze Behavior.** Appendix D provides additional information on gaze behavior, including detailed plots of average gains per trial as well as saccades and heatmaps of where gaze hit in the plane of the stimulus.
(PDF)

**S5 Appendix. Appendix E: Deviations from the Pre-Registration.** Appendix E contains all deviations from the pre-registration
(PDF)

## Author contributions

**Conceptualization:** Björn Jörges, Laurence R. Harris.

**Data curation:** Björn Jörges.

**Formal analysis:** Björn Jörges.

**Funding acquisition:** Björn Jörges, Laurence R. Harris.

**Investigation:** Björn Jörges.

**Methodology:** Björn Jörges.

**Project administration:** Björn Jörges.

**Resources:** Björn Jörges.

**Software:** Björn Jörges.

**Supervision:** Laurence R. Harris.

**Validation:** Björn Jörges.

**Visualization:** Björn Jörges.

**Writing – original draft:** Björn Jörges.

**Writing – review & editing:** Björn Jörges, Laurence R. Harris.

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
