## [Decision Letter · Decision Letter 0]

1 Sep 2025

Dear Dr. Jörges,

Thank you for submitting your manuscript to PLOS ONE. After careful consideration, we feel that it has merit but does not fully meet PLOS ONE’s publication criteria as it currently stands. Therefore, we invite you to submit a revised version of the manuscript that addresses the points raised during the review process.

We look forward to receiving your revised manuscript.

Kind regards,

Nick Fogt

Academic Editor

PLOS ONE

Journal Requirements:

“We thank the Canadian Space Agency for partial funding of this project. LRH was funded by Discovery Grant RGPIN-2020-06093 from the Natural Sciences and Engineering Research Council of Canada. BJ was funded by a Postdoctoral Transition Fellowship from the Centre for Vision Research at York University, Toronto.”

“We thank the Canadian Space Agency for partial funding of this project. LRH was funded by Discovery Grant RGPIN-2020-06093 from the Natural Sciences and Engineering Research Council of Canada. BJ was funded by a Postdoctoral Transition Fellowship from the Centre for Vision Research at York University, Toronto.”

“We thank the Canadian Space Agency for partial funding of this project. LRH was funded by Discovery Grant RGPIN-2020-06093 from the Natural Sciences and Engineering Research Council of Canada. BJ was funded by a Postdoctoral Transition Fellowship from the Centre for Vision Research at York University, Toronto.”

“We thank the Canadian Space Agency for partial funding of this project. LRH was funded by Discovery Grant RGPIN-2020-06093 from the Natural Sciences and Engineering Research Council of Canada. BJ was funded by a Postdoctoral Transition Fellowship from the Centre for Vision Research at York University, Toronto.”

6. We note that Figure 1 in your submission contain copyrighted image. All PLOS content is published under the Creative Commons Attribution License (CC BY 4.0), which means that the manuscript, images, and Supporting Information files will be freely available online, and any third party is permitted to access, download, copy, distribute, and use these materials in any way, even commercially, with proper attribution. For more information, see our copyright guidelines: http://journals.plos.org/plosone/s/licenses-and-copyright.

Additional Editor Comments:

Both reviewers have expressed serious concerns about the manuscript and it is unclear whether these concerns can be addressed. Please see specific comments below.

Reviewer #1:

The comments from reviewer #1 regarding more detailed information on the nature of the eye movements seen in the different conditions should be extensively addressed (recognizing both that eye movements could not be recorded in one condition and the authors went to significant lengths to ensure that subjects made the proper eye movements).

Reviewer #2:

Reviewer #2 asks for more detailed information on the stimulus, poses questions about the apparent mismatches between the pre-registration document and the study as performed, and asks questions around the power analysis. Most significantly, reviewer #2 suggests that the current paper may not do enough to address differences from Dichgans (1975) study. The same question could apply regarding the differentiation of the current study from the results of Freeman's 1998 study. The authors have made an attempt to put the results of the current study in the context of these previous studies, but more elaboration is needed.

Reviewers' comments:

Reviewer's Responses to Questions

**Comments to the Author**

1. Is the manuscript technically sound, and do the data support the conclusions?

Reviewer #1: Yes

Reviewer #2: Partly

2. Has the statistical analysis been performed appropriately and rigorously?

Reviewer #1: Yes

Reviewer #2: Yes

3. Have the authors made all data underlying the findings in their manuscript fully available?

Reviewer #1: No

Reviewer #2: Yes

4. Is the manuscript presented in an intelligible fashion and written in standard English?

Reviewer #1: Yes

Reviewer #2: Yes

Reviewer #1: This study focused on the Auber-Fleischl phenomenon, in which humans tend to perceive the speed of a moving object as slower when tracking it with their gaze than when their gaze is fixed. Previous experiments on the Auber-Fleischl effect have generally used targets moving against a blank background, but in this experiment, we hypothesized that in the case of moving objects in the real world, the relative movement of the object and the background provides additional clues, and that the visual system may compensate for the Auber-Fleischl phenomenon. Therefore, the authors conducted a two-interval forced-choice task comparing the speed of a single sphere with that of a cloud of spheres. The authors demonstrated that there was no evidence of the Aubert-Fleischl effect, specifically a reduction in the underestimation of speed during pursuit and fixation, in either environment. These results did not support the Bayesian model based on the idea that accuracy decline during tracking makes judgments vulnerable to the prior distribution of slow motion.

The hypotheses and results of this study are interesting, but there are several questions regarding whether the results obtained from the experimental methods and analysis of this study lead to the conclusions of this study, and several revisions are necessary to draw objective conclusions based on the results of this study. Specific questions are listed below.

Specific comments:

1) First, regarding the effect of eye movements on the results of this experiment, the generalized linear mixed model includes eye movement velocity as a fixed effect, and comparisons are also made between trials where eye velocity is above and below the reference value. The lack of eye movement velocity data and information on the gain of smooth pursuit or the velocity of catch-up saccades in response to stimulus velocity makes it difficult to interpret the comparison of velocity perception between fixation and pursuit conditions. For example, it is unclear whether OKR is elicited by visual motion from the stimulus during the fixation condition. It is also necessary to consider the magnitude of retinal slip based on the pursuit gain.

2) Among the target movement speeds used in this study, it seems possible to maintain a certain degree of pursuit gain at 15 deg/s, but at speeds of 30 and 40 deg/s, it is difficult to maintain high gain, so the gain is likely to be low and many catch-up saccades are likely to be mixed in. In such cases, it is easy to imagine that significant retinal slip will occur and that the frequency of catch-up saccades will have a significant impact on perception. In other words, when comparing fixed and pursuit conditions at different stimulus speeds, it is necessary to consider the accuracy of eye movements. Furthermore, many previous studies that demonstrated the overt Fleischl effect used relatively slow stimulus speeds within the range where pursuit gain could be maintained (e.g., less than 15 deg/s). Therefore, when comparing the results of this study, which used relatively fast conditions, with previous studies, it is necessary to consider pursuit gain. When making such comparisons, unless the dynamics of eye movements are clearly demonstrated, it is difficult for readers to understand the factors that led to these results.

3) In addition, when the background is textured, if the subject can ignore the texture information of the background to a certain extent, the pursuit gain is maintained, but if attention is directed to the background, the gain is expected to decrease further. Therefore, it is necessary to objectively evaluate the eye movements of the subjects in comparison with the JND shown in this result.

4) The results of this study indicate that the perceived speed in the pursuit condition was numerically higher than that in the fixation condition in both the relative motion condition and the no relative motion condition. This shows the opposite effect to that reported in the previous study by Aubert-Fleischl, but it is probably the result of significant retinal slip and many catch-up saccades occurring in the pursuit condition. In particular, in the tracking condition, information from eye movement reference copies, proprioception, and retinal slip is combined to estimate motion. However, each of these information sources contains inherent noise, and one possible cause of unstable velocity estimation during integration is the presence of this noise. Can the inherent noise be estimated from the eye movement data in this experiment?

5) Regarding the moderate evidence that there was no difference in PSE between trials that met the eye movement criteria and those that did not, what eye movement dynamics were exhibited in trials that did not meet the criteria? For example, why is the eye velocity so slow in cases that do not meet the criteria? Without this information, it is difficult for readers to interpret the results.

Reviewer #2: The basic idea of the manuscript is reasonable and the experiment seems to be competently done. Still, I am puzzled by many aspects of the manuscript. Maybe a revision could make a difference; I am not sure.

Things that concern me include:

1. I don't understand what the sphere cloud looks like. I tried to watch the video links to OSF, but the videos don't play. (As I type of my review, I realized that if I download the mpg files, I can play them locally, but they don't play in my web browser. That helps.) I do think the text should better explain the stimuli.

2. The pre-registration document is not much of a pre-registration. It feels more like a draft of the manuscript that was written after gathering pilot data. It is similar to a pre-registration document, but not quite. The outlier criteria different from what is in the pre-registration document, without comment or justification.

3. When removing the texture wall, the authors used black, which then had the impact of preventing measure of eye movements. Why not just use a non-textured (e.g., white) wall?

4. Many of the results and conclusions seem similar to those of Dichgans et al. (1975). Why did this current study need to be run?

5. I find it annoying that the multiple links to the OSF are to different projects. Why not put everything together in one project? (Just to be clear, I very much appreciate the authors making everything available, but it confusing to distribute things across different projects.)

6. The power analysis is weird, and possibly unnecessary. The authors correctly note that trying to do a Bayesian "power analysis" is computationally challenging, but that seems like little motivation to just do a frequentist power analysis. It possibly unnecessary because a Bayesian analysis doesn't even really have a concept of Type I or II error. You could have just added participants until you got strong enough result (one way or another).

7. The manuscript has quite a few minor grammar issues. I marked up the ones I noticed (along with some additional comments/corrections) in the attached PDF.

**Do you want your identity to be public for this peer review?** For information about this choice, including consent withdrawal, please see our Privacy Policy

Reviewer #1: No

Reviewer #2: No

---

## [Author Response · Author response to Decision Letter 1]

29 Sep 2025

Please see the response to the reviewers letter that we have uploaded separately

---

## [Decision Letter · Decision Letter 1]

14 Oct 2025

Dear Dr. Jörges,

Thank you for submitting your manuscript to PLOS ONE. After careful consideration, we feel that it has merit but does not fully meet PLOS ONE’s publication criteria as it currently stands. Therefore, we invite you to submit a revised version of the manuscript that addresses the points raised during the review process.

We look forward to receiving your revised manuscript.

Kind regards,

Nick Fogt

Academic Editor

PLOS ONE

Journal Requirements:

Additional Editor Comments:

Thank you for your responses to the reviewer comments. Please respond to the latest reviewer comments.

Specifically:

1. Please address in detail what is happening with the smooth pursuit gains. Reviewer #1 points out that they are quite high and perhaps unusual. Please verify that they are correct, and perhaps acknowledge (if appropriate) that the problems identified by the reviewer are indeed limitations. Please address whether such a limitation reduces confidence in the interpretation of the results.

2. Regarding the pre-registration issue, please describe in detail in the paper why there was a modification in the pre-registration (after submission of the paper?) and why such a modification was done/necessary.

3. Please go over the marked up manuscript provided by Reviewer #2 and address the grammatical errors.

Reviewers' comments:

Reviewer's Responses to Questions

**Comments to the Author**

Reviewer #1: (No Response)

Reviewer #2: (No Response)

2. Is the manuscript technically sound, and do the data support the conclusions?

Reviewer #1: No

Reviewer #2: Yes

3. Has the statistical analysis been performed appropriately and rigorously?

Reviewer #1: (No Response)

Reviewer #2: Yes

4. Have the authors made all data underlying the findings in their manuscript fully available?

Reviewer #1: Yes

Reviewer #2: Yes

5. Is the manuscript presented in an intelligible fashion and written in standard English?

Reviewer #1: Yes

Reviewer #2: No

Reviewer #1: The authors have responded to my previous comment by presenting additional data and figures. However, since this study was unable to accurately record eye velocity during tracking, it is still questionable whether the study's conclusions can be drawn for the following reasons.

First, regarding smooth pursuit gain, it is difficult to determine whether the eye movement data presented in these results is accurate. For example, researchers who have accurately recorded human smooth pursuit understand that tracking speeds above 30 deg/s with a gain of 0.98 or higher are nearly impossible. Furthermore, it is a well-known fact that even if the eye velocity momentarily reaches the stimulus velocity during pursuit of a constant-speed stimulus, maintaining a steady velocity is difficult. For example, the following paper (Behling S, Lisberger SG. J Neurophysiol. 2023, 130(3):652-670, Figure 2) presents eye velocity trajectories for constant-speed stimuli. Maintaining velocity becomes even more difficult for high-speed stimuli. As demonstrated by the authors in this study, it is objectively difficult to believe that pursuit can be tracked with a gain of 0.98 or higher for stimuli of 30 deg/s and 45 deg/s. Furthermore, Figure A7A presents results where pursuit gain significantly exceeds 1.0, but it is generally considered nearly impossible for pursuit velocity to overshoot stimulus velocity. In addition, while overshoot may occur during the initial phase of pursuit, it does not persist continuously. This means that to clarify the velocity perception obtained in the study, one must carefully examine the effects of eye movement and retinal slip velocity; otherwise, one encounters a barrier preventing understanding of the essence.

Therefore, it is essential to first prove whether the smooth pursuit gain demonstrated in this study is objectively correct when compared to prior research. Alternatively, if this proves difficult, it should be presented as a limitation of this study. Conclusions should then be drawn under the premise that the velocity perception obtained in this study may be influenced by eye movements and retinal slip velocity.

Reviewer #2: The revised manuscript is much improved. In particular, small changes to the text make it easier to understand the stimuli and task. Still, there are a few issues.

1) There are still quite a few grammatical issues. Some of these are carried over from my previous review. I, again, marked up a copy of the manuscript with some corrections and comments.

2) The pre-registration document at the OSF is a bit problematic. There are some deviations in the manuscript that are not explained. Worse, the OSF site indicates that the pre-registration document was modified September 25, 2025, which means it was modified since the submission of the original manuscript. If so, then it is no longer a pre-registration document. As a result, I think it is wrong for the manuscript to refer to analyses being preregistered. Maybe there were, but readers cannot verify the situation with the online materials.

**Do you want your identity to be public for this peer review?** For information about this choice, including consent withdrawal, please see our Privacy Policy

Reviewer #1: No

Reviewer #2: No

---

## [Author Response · Author response to Decision Letter 2]

29 Oct 2025

Please see the attached "Reponse to the reviewers" file.

---

## [Editor Report · Decision Letter 2]

4 Nov 2025

Failure to Replicate the Aubert-Fleischl Effect

PONE-D-25-21124R2

Dear Dr. Jörges,

We’re pleased to inform you that your manuscript has been judged scientifically suitable for publication and will be formally accepted for publication once it meets all outstanding technical requirements.

Kind regards,

Nick Fogt

Academic Editor

PLOS ONE

Additional Editor Comments (optional):

Thank you for your thorough responses to the reviewer comments.
---

## [Editor Report · Acceptance letter]

PONE-D-25-21124R2

PLOS ONE

Dear Dr. Jörges,

I'm pleased to inform you that your manuscript has been deemed suitable for publication in PLOS ONE. Congratulations! Your manuscript is now being handed over to our production team.

Kind regards,

on behalf of

Dr. Nick Fogt

Academic Editor

PLOS ONE